# Modelling Human Visual Motion Processing with Trainable Motion Energy Sensing and a Self-attention Network

**Zitang Sun**[1]   **Yen-Ju Chen**[1]   **Yung-Hao Yang**[1]   **Shin'ya Nishida**[12*]
[1] Graduate School of Informatics, Kyoto University
[2] NTT Communication Science Laboratories, Nippon Telegraph and Telephone Corporation
`{sun.zitang.73u, chen.yenju.44z, yang.yunghao.8v}`
@st.kyoto-u.ac.jp; nishida.shinya.2x@kyoto-u.ac.jp

## Abstract

Visual motion processing is essential for humans to perceive and interact with dynamic environments. Despite extensive research in cognitive neuroscience, image-computable models that can extract informative motion flow from natural scenes in a manner consistent with human visual processing have yet to be established. Meanwhile, recent advancements in computer vision (CV), propelled by deep learning, have led to significant progress in optical flow estimation, a task closely related to motion perception. Here we propose an image-computable model of human motion perception by bridging the gap between biological and CV models. Specifically, we introduce a novel two-stages approach that combines trainable motion energy sensing with a recurrent self-attention network for adaptive motion integration and segregation. This model architecture aims to capture the computations in V1-MT, the core structure for motion perception in the biological visual system, while providing the ability to derive informative motion flow for a wide range of stimuli, including complex natural scenes. In silico neurophysiology reveals that our model's unit responses are similar to mammalian neural recordings regarding motion pooling and speed tuning. The proposed model can also replicate human responses to a range of stimuli examined in past psychophysical studies. The experimental results on the Sintel benchmark demonstrate that our model predicts human responses better than the ground truth, whereas the state-of-the-art CV models show the opposite. Our study provides a computational architecture consistent with human visual motion processing, although the physiological correspondence may not be exact. [1]

## 1   Introduction

Visual motion perception is essential not only for humans and animals to perceive and interact with the world but also for artificial agents to process various dynamic visual tasks. As such, visual motion estimation has been extensively studied by both biological vision and computer vision research communities. The key issue lies in estimating optical flow, an array of instantaneous image motion vectors [1, 2].

Vision science has revealed that optical flow estimation in the human visual system (HVS) is primarily served by a pathway that includes the primary visual cortex (V1 [3, 4]) and middle temporal (MT [5, 6]) area. A significant portion of neurons in area V1 are sensitive to the direction of local motion, while those in area MT integrate and segregate local motion signals for global flow interpretation. The MT process also helps to overcome the aperture problem[7]. Several computational models of area MT have been proposed [8, 9], but mechanisms that can fully encapsulate a variety

---

[1]Project Website: Click here

37th Conference on Neural Information Processing Systems (NeurIPS 2023).

of motion integration abilities of MT neurons and/or human perception have yet to be developed [10]. Furthermore, previous studies tested the models using only simple artificial stimuli. It remains challenging to create a human-like optical flow extraction mechanism that can derive informative motion flow for a wide range of stimuli including complex natural videos.

On the other hand, optical flow estimation has recently made remarkable progress in the field of computer vision. FlowNet[11] pioneered the use of fully convolutional neural networks for dense optical flow estimation, and various approaches based on deep neural networks (DNNs) have emerged subsequently[12–15]. Owing to the powerful representation ability of DNNs, these models outperform humans in estimating the ground-truth optical flow of natural scenes [16]. Consequently, one might expect DNN-based optical flow estimation algorithms to become promising models of human visual motion processing, similar to how ImageNet-trained DNN models provide good computational models of human object recognition[17]. However, since computer vision optical flow models are designed solely to find local image correspondences between pairs of frames on the image coordinates [11], they cannot explain systematic or adaptive deviations of human perceived motion from local ground truth (GT) [16]. Moreover, existing DNN models often exhibit instability when dealing with non-textured stimuli commonly used in vision science [18].

In this study, we leveraged the flexibility of DNNs to construct an image-computational model of human motion processing. While recent studies successfully used DNNs to elaborate the understanding of the neural mechanism of visual motion processing[19–23], we aimed to make a model that can explain a broader range of physiological and psychophysical phenomena, including those whose neural mechanisms are not yet clear. From an engineering standpoint, we aimed to make a human-aligned optic flow algorithm while maintaining a flow estimation capability comparable to the state-of-the-art (SOTA) CV models. Our model extracts dense informative motion flows for a wide range of inputs in a way consistent with physiologically measured neural responses and psychophysically measured human motion perception. It consisted of two stages. The first stage, which mimicked the function of V1, comprised of neurons with multi-scale spatiotemporal filters to extract local motion energy. Unlike previous models of V1 [24, 8, 25], the filter tunings were learnable to fit natural optic flow computation. The second stage, which mimicked the function of MT recurrently integrated local motion signals, and solved the aperture problem. We constructed an undirected fully connected graph from the map of local motion energy, and used the attention mechanism [26, 27] for adaptive global motion integration and segregation.

We evaluated the performance of our model from several aspects. In in-silico neurophysiology, we found that our model's neurons exhibited direction and speed tunings similar to those observed in mammalian physiological recordings in V1 and MT [28, 29]. In simulations of psychophysical findings, our model showed good generalization from simple artificial stimuli (e.g., drifting Gabor) to complex naturalistic scenes. Our model produced human-like responses for several conventional motion stimuli and illusions, including global motion pooling and the barber-pole illusion. Furthermore, the mode's response to natural scenes was closer to that of humans in comparison to other computer vision models. Our two-stages model provides a computational architecture consistent with human visual motion processing, although the physiological correspondence may not be exact. This achievement is not only valuable in terms of neuroscientific understanding of human visual computation but also for the development of human-aligned machine vision that stably recognizes the world as humans do.

## 2 Molding of two-stages motion perception system

### 2.1 First-stage: Local Motion Energy Computation

**Spatiotemporal separable Gabor filter:** Since we aimed at an image-computable model, the input is a sequence of grayscale images $\mathbf{S}(\mathbf{p}, \mathbf{t})$ for all spatial positions $\mathbf{p} = (x, y)$ within the image domain $\mathbf{\Omega}$ and for all times $\mathbf{t} > \mathbf{0}$. The goal of the first-stage neuron is to capture local motion energy at a specific spatiotemporal frequency, which is associated with the function of a direction-selective neuron in the V1 cortex. The responses of neurons can be modeled as 3D Gabor filters [30, 31]. Gabor filters are known to be optimal in the sense that they achieve maximal resolution in both the spatiotemporal and associated frequency domains[32]. To save computational complexity, we decomposed 3D spatiotemporal filters into filters separable in space and time. The spatial component is described by 2D Gabor filters $\mathcal{G}(\cdot)$, and the temporal component $\mathcal{T}(\cdot)$ is described

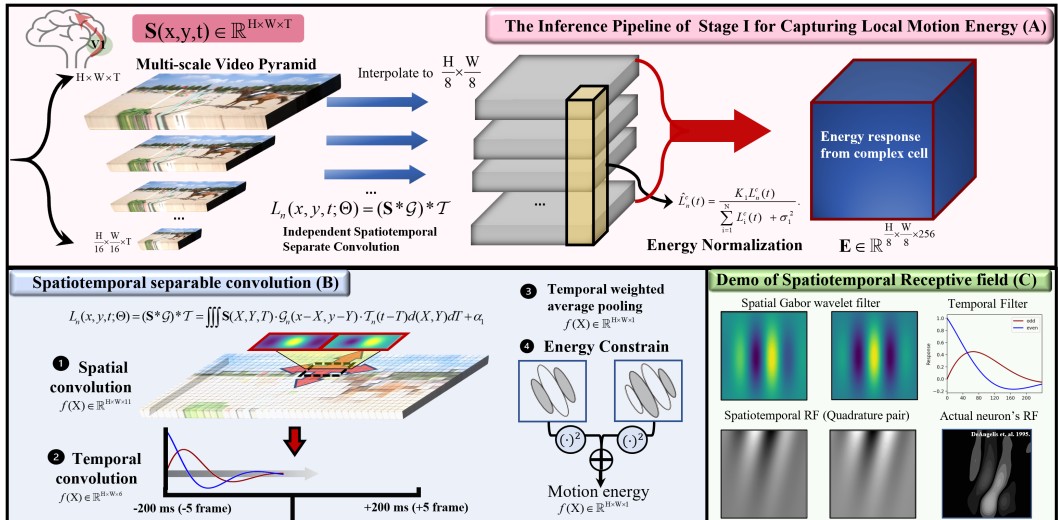

Figure 1: *Molding of motion perception system, Stage I.* **(A)**: The first stage is built using a group of trainable motion energy units to capture local motion energy; **(B)**: Motion energy calculation based on spatiotemporal separable filters, a sub-block of **(A)**; **(C)**: Demo of spatiotemporal separable filters, a sub-block of **(B)**.

by a 1D sinusoidal function with exponential decay. Specifically, given $x' = x\cos\theta + y\sin\theta$ and $y' = -x\sin\theta + y\cos\theta$ within the receptive field, the impulse responses of spatial and temporal complex filters are defined as:

$$\begin{cases} \mathcal{G}(x, y; f_s, \theta, \sigma, \gamma) = \exp\left(-\frac{x'^2 + \gamma^2 y'^2}{2\sigma^2}\right)\exp\left(i\left(2\pi f_s x'\right)\right), s.t. \{x, y \,|\, (x^2 + y^2 \le R^2)\} \\ \mathcal{T}(t; f_t, \tau) = \exp\left(-\frac{t}{\tau}\right)\exp\left(2\pi i\left(f_t t\right)\right), s.t. \{t \,|\, 0 \le t < T\} \end{cases}$$
(1)

The parameters in red are designed to be trained to fit the dataset, where $f_s$ and $f_t$ denote the spatiotemporal frequency tunings of the filter with the preferred speed $v \propto \frac{f_t}{f_s}$, and $\theta$ determines the preferred moving orientation; $\sigma$ and $\gamma$ control the shape of the Gabor filter, and $\tau$ controls the degree of attenuation of the temporal impulse response. All parameters are subject to certain numerical constraints, such as $\theta$ being limited to $[0, 2\pi)$ to avoid redundancy; $f_s$ and $f_t$ are limited to less than 0.25 pixels per frame to avoid spectrum aliasing, etc. The response of a simple direction-selective cell $L_n$ to a video stimuli $\mathbf{S}(\mathbf{p}, \mathbf{t})$ can be computed via separate convolutions:

$$L_n(x, y, t; \Theta) = (\mathbf{S} * \mathcal{G}) * \mathcal{T} = \iiint \mathbf{S}(\mathcal{X}, \mathcal{Y}, \mathcal{T}) \cdot \mathcal{G}_n(x - \mathcal{X}, y - \mathcal{Y}) \cdot \mathcal{T}_n(t - \mathcal{T}) d(\mathcal{X}, \mathcal{Y}) d\mathcal{T} + \alpha_1 \quad (2)$$

where $\alpha_1$ can be learned as spontaneous firing rates. Further, local motion energy is captured by a phase-insensitive complex cell in the V1 cortex, which computes the squared summation of the response from a pair of simple V1 cells with approximately orthogonal spatiotemporal receptive fields[24]. We denote the pair of orthogonal (even and odd) simple V1 cells as:

$$\begin{cases} L_n^o(x, y, t; \Theta) = (\mathbf{S} * \mathrm{Im}[\mathcal{G}]) * \mathrm{Re}[\mathcal{T}] + (\mathbf{S} * \mathrm{Im}[\mathcal{G}]) * \mathrm{Im}[\mathcal{T}] \\ L_n^e(x, y, t; \Theta) = (\mathbf{S} * \mathrm{Re}[\mathcal{G}]) * \mathrm{Re}[\mathcal{T}] - (\mathbf{S} * \mathrm{Im}[\mathcal{G}]) * \mathrm{Im}[\mathcal{T}], \end{cases}$$
(3)

where $\mathrm{Re}(\cdot)$ and $\mathrm{Im}(\cdot)$ extract the real and imaginary parts of a complex number; $*$ denotes convolution operations. Then, the response of a complex cell $L_n^c$ is obtained from a combination of the quadrature pair of the simple cells using the motion energy formulation:

$$L_n^c(x, y, t; \Theta) = (L_n^o(x, y, t; \Theta))^2 + (L_n^e(x, y, t; \Theta))^2 \quad (4)$$

**Multi-scale Wavelet Processing:** The convolution kernel of our spatial filter has a fixed size of $15 \times 15$, which imposes a physical limitation on the receptive field of each unit. To enhance the flexibility of the receptive field size, we employed a multi-scale processing strategy, as shown in Fig. 1 (A). Specifically, we constructed an image pyramid consisting of eight images scaled linearly

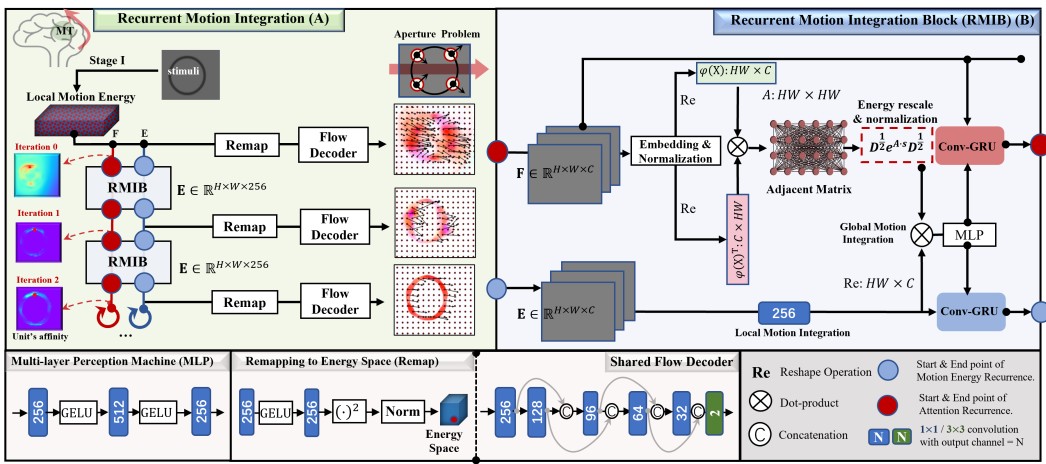

Figure 2: *Molding of motion perception system, Stage II.* **(A)**: The second Stage is constructed by using self-attention mechanism and recurrent processing to simulate the function of global motion integration and segregation. We use an identical flow decoder to visualize the dense optical flow at each iteration. **(B):** the global motion integration process based on self-attention mechanism, a sub-block of **(A)**.

from $H \times W$ to $\frac{H \times W}{16}$. In total, 256 independent complex cells are deployed across different scales, with the lower scale having a larger receptive field and a preference for a faster motion speed. This approach is computationally efficient for capturing large-scale displacements in engineering applications[33, 34], and enables the representation of different groups of cells sensitive to short- and long-distance motions [35]. The N complex cells $\{L_n^c\}_i^N$ capture motion energy on multiple scales. We applied energy normalization to each cell to ensure consistent energy levels:

$$\hat{L}_n^c(t) = \frac{K_1 L_n^c(t)}{\sum_{i=1}^{N} L_i^c(t) + \sigma_1},\tag{5}$$

where $\sigma_1$ is the semi-saturation constant of the normalization, and $K_1 > 0$ determines the maximum attainable response. We interpret the response, denoted by $\hat{L}_n(t)$, as the model equivalent of a post-stimulus time histogram (PSTH), which is a measure of the neuron's firing rate. Physiologically, such responses could be computed via inhibitory feedback mechanisms[36, 37]. Considering the spatial arrangement of images, it is expected that motion energy responses should exist at each spatial location generated by the same complex cell groups. Bilinear interpolation is used to resize the multi-scale motion energy into the same spatial size $= \frac{H \times W}{8}$. In the context of DNNs, this is mainly to balance the trade-off between the spatial resolution and computational overhead, and the final output of the first stage is a 256-channel feature map $\mathbf{E} \in \mathbb{R}^{\frac{H}{8} \times \frac{W}{8}}$ that captures the underlying local motion energy, which partially characterizes the cellular patterns of the V1 cortex in a computational manner[24].

## 2.2 Second stage: Global Motion Integration and Segregation

The receptive field size of first-stage neurons limits their ability to capture only local motion. Advanced spatial integration is an essential requirement for a motion perception system to solve the aperture problem[7]. Spatial integration could involve several mechanisms, such as object recognition and segmentation[38], depth estimation[39], contour structure inference[40], etc., which rely on substantial prior information that may go beyond the capabilities of classical modeling methods. DNNs, with their massive number of parameters and flexibility, offer a suitable solution. However, the spatial integration of local motion requires more flexible connectivity relations than what can be achieved by general convolutions, which are limited by their local receptive field[41]. To overcome this limitation, we propose a computational model based on the attention mechanism and recurrent processing to model the function of motion integration.

**Constructing a Graph Structure:** First, to establish more flexible connections between cells, we discard the Euclidean space structure within the image and construct topological spaces with an

undirected weighted graph $\mathbf{G} = \{\mathbf{V}, \mathbf{A}\}$, where $\mathbf{V}$ is the set of nodes; $\mathbf{A}$ represents the adjacent matrix based on the graph. Each spatial location $p(i, j)$ is treated as a node, and the feature of each node is derived from the whole set of local motion energies $\mathbf{E}(i, j) = \{\hat{L}_n^c(t)\}_{i=1}^{256}$. The connection between any pair of nodes is computed using a specific distance metric, and strong connection relationships are formed between nodes whose local motion energy patterns are similar.

Specifically, given a reshaped feature map $\mathbf{F} \in \mathbb{R}^{HW \times 256}$, the features at each of its locations are first locally embedded into the graph space, as $\varphi(\mathbf{F}) = \text{GELU}(\mathbf{F} \cdot \mathbf{W}_{\Theta 1}) \cdot \mathbf{W}_{\Theta 2}$, where $\mathbf{W}_\Theta \in \mathbb{R}^{256 \times 256}$ is a group of trainable parameters. The distance between any pair of nodes $(i, j)$ is calculated by the cosine similarity, which is similar to the self-attention mechanism in current transformer structures[26, 42, 27]. We employ the adjacency matrix $\mathbf{A} \in \mathbb{R}^{HW \times HW}$ to represent the connectivity of the whole topological space, and $\mathbf{A}$ is a symmetric semi-positive definite matrix defined as:

$$\mathbf{A}(i, j) = \mathbf{A}(j, i) = \frac{\varphi(\mathbf{F})_i \cdot \varphi(\mathbf{F})_j}{\|\varphi(\mathbf{F})_i\| \|\varphi(\mathbf{F})_j\|}. \tag{6}$$

We perform exponential scaling of the connections between graphs using the matrix $\mathbf{A}$, given by $\exp(\mathbf{A}s)$, where $s$ is a learnable scalar restricted to the range $(0,10)$ to avoid overflow. The smaller $s$, the smoother the connections across nodes and vice versa. Finally, a symmetric normalization operation is utilized to balance the energy, given by $\mathbf{A} := \mathbf{D}^{-\frac{1}{2}} \exp(s\mathbf{A}) \mathbf{D}^{-\frac{1}{2}}$, where $\mathbf{D}$ is the degree matrix with $\mathbf{D} = \text{diag}\left(\left\{\sum_j \exp(s\mathbf{A}_{i,j})\right\}_{i-1}^n\right)$. As such, an energy-normalized undirected graph structure is constructed, as illustrated in the top side of Fig. 2 (B). Intuitively, this adjacency matrix represents the neuron's affinity or connectivity within the space, with strong and global connections built across neurons with related motion responses.

**Recurrent Integration Processing:** Recurrent neural networks (RNNs) are often used to simulate neurons in the brain, as they are flexible in modelling temporal dependencies and feedback loops, which are fundamental aspects of neural processing in the brain[43]. We use a recurrent network, rather than multiple feedforward blocks, to simulate the process of local motion signals being gradually integrated into MT and eventually converging to a stable state.

As shown in Fig. 2 (A), the local motion energy from the first stage is divided into two recurrent streamlines. One is the motion energy $\mathbf{E} \in \mathbb{R}^{H \times W \times 256}$, which is continuously updated in the loop, while the other is embedded in the attention space to generate the graph adjacency matrix to control motion integration, denoted as $\mathbf{F} \in \mathbb{R}^{H \times W \times 256}$. In each iteration, the adjacency matrix is first constructed using $\mathbf{F}$. Subsequent motion integration is achieved through a simple matrix multiplication, which is computationally similar to the information propagation mechanism in transformers[26, 42] and can also be considered a simplified version of graph convolution[44]. The integrated motion information is passed through two independent Conv-GRU blocks to update the motion energy $\mathbf{E}$ and feature $\mathbf{F}$, respectively. The Conv-GRU represents a gated recurrent unit[45] implemented in a convolutional manner, and we adopt a spatio-temporal separable approach following RAFT[14]. The motion integration process approximates the ideal final convergence state of the motion energy $\mathbf{E}_k \to \mathbf{E}^*$ through recurrent iteration.

**Decoding the Motion Flow:** We adopt the same strategy of decoding the 2D optical flow in each iteration. Initially, the integrated motion is $\mathbf{E}$ projected back to the energy space with positive value using a square operation, followed by an energy normalization operation: $\hat{\mathbf{E}}(i, j) = \{K_2 \mathbf{E}^2(i, j)\}/\{\sum_{i,j}^{HW} \mathbf{E}^2(i, j) + \sigma_2{}^2\}$. The resulting response $\hat{\mathbf{E}} \in \mathbb{R}^{H \times W \times 256}$ is interpreted as a post-stimulus time histogram and the motion energy is constrained to the same energy space as the local motion energy from stage I, as we further design an identical flow decoder to project the energy of each spatial location into the optical flow. The structure of the flow decoder is a nonlinear mapping process consisting of multiple $1 \times 1$ convolution blocks with residual connections, which are referred to in several current advanced optical flow estimation models[33, 46]. Additionally, a convex upsampling strategy[14] is employed to restore the optical flow's original resolution. The entire architecture of stage II is illustrated in Fig. 2 (A, B).

## 2.3 Training Strategy

Current methods for estimating optical flow using deep neural networks (DNNs) can be categorized as unsupervised/self-supervised and supervised learning approaches. While unsupervised learn-

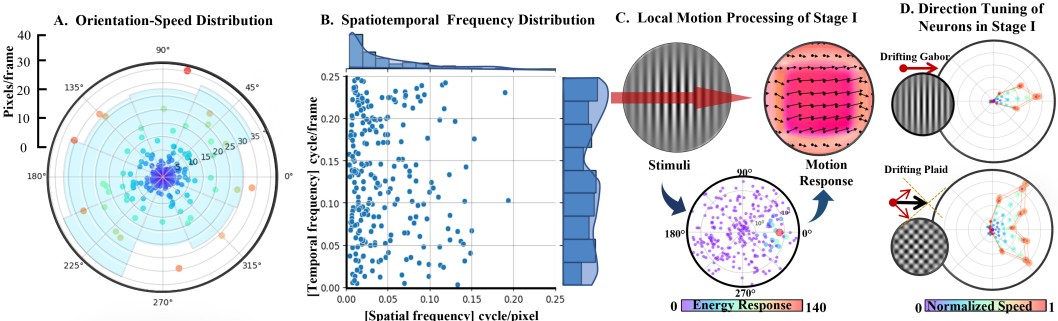

Figure 3: *Properties of trained complex cells in stage I.* **(A)**: Orientation-speed distribution of trained motion energy units. **(B)**: Spatiotemporal frequency tuning distribution of trained motion energy units. **(C), (D)**: The ability to tune simple motions with a single spatiotemporal frequency component has emerged in the stage I, exemplified by direction tuning of drifting Gabor stimuli.

ing methods are intuitively similar to creatures' interaction with the world, most current methods based on differentiable image warping[47–49] still try to approximate physical motion ground truth. Therefore, we adopt a supervised learning approach in this work, which is more straightforward as recent research suggests that human perception of motion is reasonably similar to physical GT[16]. However, our primary goal lies in evaluating how well the model approximates human motion perception rather than its accuracy in predicting GT. To train and evaluate the model, we construct a dataset containing various natural and artificial motion scenes. Specifically, we incorporate the Sintel benchmark[50], the DAVIS[51] dataset with pseudo-labels generated by FlowFormer[52], as well as self-created multi-frame datasets with non-texture motions and drifting grating motion. Including simple motion stimuli and drifting gratings allows the model to generalize under different non-texture conditions while providing a potential slow-world Bayesian prior[9]. The model is first pre-trained with simple motion and subsequently fine-tuned on complex natural scenes to facilitate convergence[12]. More specific training details can be found in the supplementary materials.

## 3   Analyses

Fig. 3 shows the distribution of trained parameters in the first stage: (A) presents the distribution of velocity and orientation of the units; (B) displays the spatiotemporal frequency tuning of the units; (C) demonstrates that the complex cells in the first stage are capable of handling single-frequency component motions such as drifting gratings. The design of the trainable motion energy unit allows for optimal fitting to the flow statistics of the dataset. Although the distribution of the trained parameters appears to lack specific characteristics other than uniformity, it does reflect the effect of training. To validate the effectiveness of training the motion energy units, we conducted experiments with a fixed tuning parameter design using a uniform distribution sampled at equal intervals in terms of spatiotemporal frequency and orientation. The results showed that stage I without the fitting function significantly degraded the model's ability to estimate motion. (See **Our-fixed** in Table 1 for details).

The following three parts are conducted for analysis: 1) In silico neurophysiological test of the activation pattern of the units; 2) Psychophysical stimulus tests comparing human perception and model response; 3) Natural scene test of the generalizability of the model in complicated scenarios.

### 3.1   In silico Neurophysiological Study

**Directional Tuning:** Some V1 and MT neurons respond selectively to a specific range of motion directions. To investigate the directional tuning of the units in our model, we utilized in silico neurophysiology to measure the activation patterns of 256 units in response to drifting Gabor and plaid stimuli. A plaid consists of two superimposed drifting Gabors, as illustrated in Fig. 4 (C). Analysis of the data revealed three distinct groups of units based on their partial correlation to Gabor and plaid stimuli: 1) Component cells, which always respond to the direction of a Gabor component; 2) Pattern cells, which respond to the integrated (coherent) motion direction of plaid; and 3) Un-

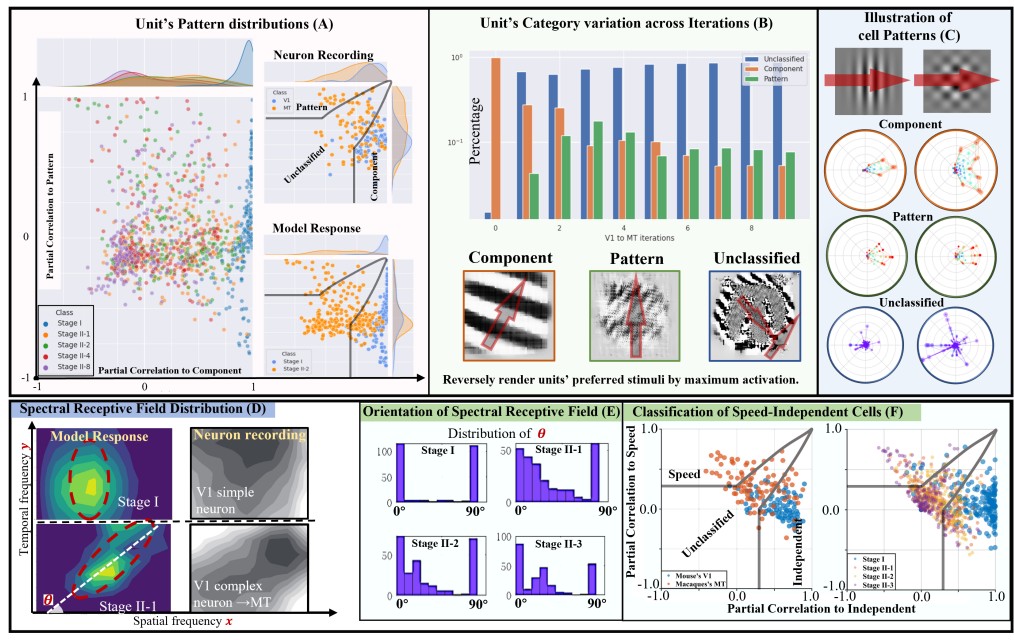

Figure 4: *In Silico Neurophysiological Test*. (**A, B, C**): Analysis of the direction tuning of different groups of neurons for 1D and 2D stimuli; Stage II-$N$ means the result from iteration $N$th in the stage II; (**D, E, F**): Analysis of the spectral receptive field and speed tuning of different groups of neurons. The neural distribution in (**A**) is redrawn from [29]. Neurons' spectral receptive fields in (**D**) are redrawn from [53]. Two neurons distributions in (**F**) are separately taken from [54] and [55].

classified cells, which do not show a clear preference for either component or pattern motions, as illustrated in Fig. 4 (C). The distribution of these cell types is not uniform, with component cells being more commonly found in the primary visual cortex (V1), while pattern cells are more often observed in the MT and MST regions[**?** ]. In agreement with this, as illustrated in Fig. 4 (A) (B), our model demonstrates that in the first stage, most units tend to be component cells, whereas the number of pattern and unclassified cells increases in the second stage. In addition, we employed the maximizing activation method [56] to reversely render their cellular preferences, showing that the second-stage unclassified units respond to more complicated motion patterns consisting of both central and background motions, as presented at the bottom of Fig. 4 (B). This suggests that the classical classification of motion neurons into component and pattern cells might be insufficient to characterize the motion integration properties of these neurons.

**Spectral Receptive Field and Speed Tuning:** We test the spectral (spatial-frequency-vs-temporal-frequency) receptive field of the model's units using a combination of drifting Gabors with different spatiotemporal frequencies. Two-dimensional oriented Gaussian contours are used to depict the receptive field of the 256 cells, fitted by minimizing the least square error. The results are shown in Fig 4 (E) . Visually, the distribution of the receptive field tilt angle spreads from horizontal/vertical directions to oblique directions (Fig. 4, E). This indicates that the units in the stage II have significant speed tuning compared to the stage I. Speed tuning is a characteristic of higher-order visual motion neurons[28] and is often found in the MT area[57]. This tendency can be seen from the distribution of the partial correlation between the actual receptive field and its speed prediction/independent prediction[55], as demonstrated in Fig. 4 (F). Our two-stages process shows a degree of consistency with the change in mammalian neural distribution from the V1 to MT area.

## 3.2 Psychophysical Analysis

**Fourier motion**: In the "missing fundamental illusion"[60], as shown in Fig. 5 (C), when the first spatial harmonic is removed from a square-wave grating with a quarter-cycle shift, the perceived motion direction appears reversed. Our model, whose first stage estimates motion from the Fourier

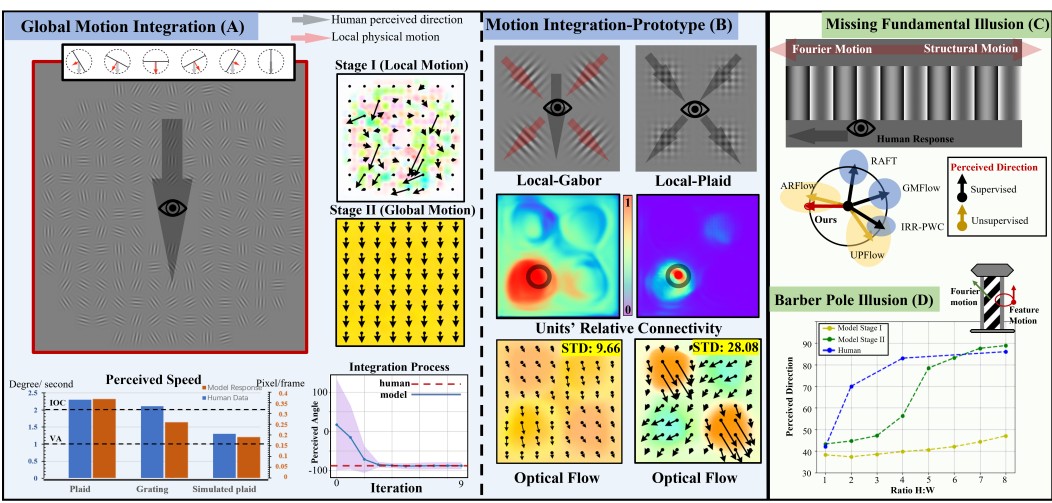

Figure 5: *Psychophysical stimulus tests compare human perception and model response.* **(A, B):** Spatial integration of 1D and 2D motions. In the stimulus panels, grey arrows indicate the direction perceived by humans, while red arrows indicate the direction of physical motion. The middle panel of **(B):** is the heat map representing the neuron's connectivity from a selected local region (circle center) to other regions, with the warmer color indicating a stronger connection. **(C):** Fourie-motion-based illusion. The size of the shaded area indicates the pixel-wise standard deviation (STD) of the response. **(D):** Barber pole illusion. The human psychophysical data in **(A)**, **(D)** were redrawn from the [58] and [59], respectively.

components[61], can predict this reversal. In contrast, computer vision models designed to infer optical flow based on structural correspondence do not exhibit this bias.

**Motion Integration:** The direction of 1D motion stimuli, such as drifting Gabors, is ambiguous due to the aperture problem. When presented alone, they are perceived to move in orthogonal directions of stripes. When superimposed with other 1D motion components in different orientations, 2D directions consistent with both components are perceived. Pattern motion neurons in MT may contribute to this phenomenon. The second stage of our model can explain this as well, as shown in Fig. 5. Psychophysics also demonstrates motion integration across space. Fig. 5 (A) shows a psychophysical stimulus consisting of drifting Gabors [58]. These Gabors have a variety of local directions and speeds, yet all of them are consistent with one global 2D motion (downward in this case). When viewed as a whole, humans do perceive coherent downward motion. Stage II of our model predicts both the perceived direction and speed of the global Gabor motion.

Fig. 5 (B) compares spatial motion integration between 1D Gabor motion (left) and 2D plaid motion (right). Humans are able to perceive global downward motion only in the former case: The heat maps depict how units with high activity establish long-distance connections to resolve the aperture problem when subjected to Gabor (ambiguous motion) stimuli. In contrast, the plaid stimuli (determined motion) suppress these long-distance connections. In the latter case, local integration of motion signals takes priority over global integration. Once the local ambiguity is resolved, the global integration process is suppressed. Our model can predict such adaptive motion pooling in human visual processing [58]. Furthermore, the barber pole illusion demonstrates how locally ambiguous 1D motion is affected by the shape of the moving area [62]. Specifically, as the height-width ratio of the visual area varies, human perception of direction shifts from oblique to vertical [59] (Fig. 5, D). Our model can predict the shift in perceived direction in stage II, showing its ability to integrate motion signals with boundary orientations. For more video demonstrations, such as the reverse phi illusion[63], please see the supplementary material.

**Comparison of Complex Natural Scenes:** The proposed model can effectively handle natural scenes, as demonstrated in Fig. 6 (A). Natural stimuli contain diverse spatiotemporal frequency components, leading to complex activation patterns in Stage I. From the decoded flow field, the motion of stage I is limited to localized areas. For example, local motion cannot be found in untextured road areas. This situation necessitates long-range interactions with the surrounding spatial context, which our recurrent integration process in stage II effectively accomplishes. It is evident from the

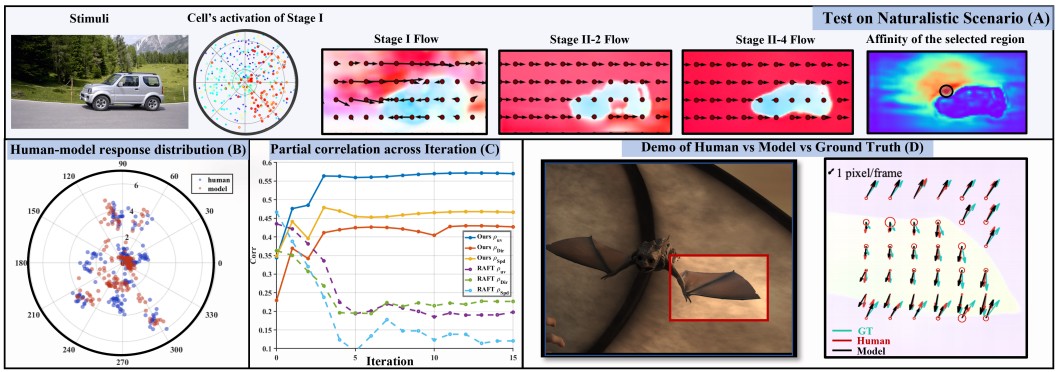

Figure 6: *Comparison of Performance on Complex Natural Scenes.* **(A)**: Our model can estimate a dense natural optical flow with the recurrent integration. **(B-D)**: Similarity of our model estimation to human perceived flow on Sintel dataset. The red circle's size in **(D)** shows how much closer our model is to the human response than GT.

affinity heat map (right side of subfigure A) that object and background areas are clearly segregated in stage II. This suggests that the integration mechanism based on attention has the potential to combine motion integration and object segmentation into a single framework. These two processes are considered highly relevant in the human visual system [64].

We used psychophysically measured optical flow from the Sintel dataset[16] as a benchmark for naturalistic scene flow perceived by humans. Our model was compared to several optical flow estimation methods used in computer vision, including classical algorithms like Farneback, as well as SOTA DNN-based models. We evaluated officially released DNN models that utilize a wide range of inference structures, such as multi-scale inference[11, 12], spatial recurrent models[14], graph reasoning[65], and vision transformers[15].

As shown in Table 1, we computed both the Pearson correlation coefficients and vector endpoint errors (EPEs) between the model prediction and human response, or ground truth (GT). Additionally, we examined the partial correlation between humans and models while controlling the impact of GT:

$$\rho_{\text{model}} = r_{\text{resp model} \cdot GT} = \frac{r_{\text{resp model}} - r_{\text{resp } GT} \cdot r_{\text{model } GT}}{\sqrt{1 - r_{\text{resp } GT}^2}\sqrt{1 - r_{\text{model } GT}^2}}, \tag{7}$$

where $r$ is the Pearson correlation. This measure is critical in validating the models' ability to capture the characteristics of the human response, as any model could appear to have a high correlation with the human response by just approximating the GT due to a high correlation between the human response and GT[16].

Quantitatively, the proposed model outperformed all compared models in terms of partial correlation. The RAFT-val in the table is the RAFT framework trained on our dataset as a validation, and indicates that our mixed training set also improves the explanatory power of the human response. Fig. 6 (C) shows that our model significantly improves the partial correlation across iterations in Stage II, indicating that the proposed recurrent motion integration architecture can generate more human-like deviations from GT, which is a trend not present in a similar recurrent network, RAFT. In Fig. 6 (D), one can directly see that our model prediction is more similar to human-perceived flow than the GT flow.

Table 1 also shows the results of three biologically-inspired models. FFV1MT[67] is a model capable of computing dense optical flow using direct decoding of the Simoncelli & Heeger V1-MT mechanism. The other two models are modified versions of MotionNet[20] and DorsalNet[23], recently proposed DNN-based models for the explanation of neural responses to visual motion stimuli. Since the original models are designed to recognize global motion only, we tested general multi-layer 3D CNN (a core component of MotionNet and DorsalNet) with residual connections, and a pre-trained DorsalNet with frozen parameters, with a linear flow decoder to compute dense flow, trained on natural dense optical flow datasets. Our model outperforms these biologically-inspired models in predicting human responses. The low performances of FFV1MT model and the modified

Table 1: **Model v.s. Human v.s. GT.** $\rho$ : Partial correlation between human & model controlling GT; $r$: Pearson correlation coefficient; $epe$: vector end-point error; $uv$, $dir$, $spd$ represent motion components in Cartesian space, direction, and speed, respectively.

| Method | $\rho_{uv}$ | $\rho_{dir}$ | $\rho_{spd}$ | v.s. Human | | | | v.s. GT | | | |
|---|---|---|---|---|---|---|---|---|---|---|---|
| | | | | $r_{uv}$ | $r_{spd}$ | $r_{dir}$ | $epe$ | $r_{uv}$ | $r_{spd}$ | $r_{dir}$ | $epe$ |
| Farneback[66] | 0.27 | 0.23 | 0.11 | 0.41 | 0.91 | 0.34 | 2.02 | 0.34 | 0.33 | 0.92 | 1.96 |
| FlowNet2.0[11] | 0.39 | 0.26 | 0.34 | 0.92 | 0.90 | 0.96 | 0.94 | 0.95 | 0.94 | 0.98 | 0.47 |
| RAFT[14] | 0.20 | 0.22 | 0.14 | 0.92 | 0.90 | 0.96 | 0.93 | 0.98 | **0.99** | **0.99** | **0.25** |
| RAFT-val | 0.43 | 0.17 | 0.42 | 0.92 | 0.89 | 0.96 | 1.01 | 0.92 | 0.89 | 0.98 | 0.69 |
| AGFlow[65] | 0.30 | 0.16 | 0.20 | 0.93 | 0.90 | 0.96 | 0.92 | 0.98 | 0.98 | 0.98 | 0.27 |
| GMFlow[15] | 0.34 | 0.32 | 0.17 | 0.91 | 0.84 | **0.96** | 1.03 | 0.93 | 0.90 | 0.97 | 0.73 |
| FlowFormer[52] | 0.36 | 0.14 | 0.32 | **0.93** | **0.91** | 0.95 | **0.90** | **0.98** | 0.97 | 0.98 | 0.42 |
| FFV1MT[67] | 0.31 | 0.16 | 0.31 | 0.83 | 0.64 | 0.92 | 1.48 | 0.59 | 0.84 | 0.94 | 1.29 |
| 3DCNN | 0.27 | 0.29 | 0.42 | 0.83 | 0.86 | 0.95 | 1.31 | 0.83 | 0.86 | 0.96 | 1.14 |
| DorsalNet[23] | 0.17 | 0.19 | -0.10 | 0.20 | -0.08 | 0.86 | 2.35 | 0.20 | -0.04 | 0.86 | 2.33 |
| **Ours-fixed** | -0.02 | 0.12 | 0.16 | 0.31 | 0.23 | 0.78 | 2.24 | 0.35 | 0.18 | 0.80 | 2.29 |
| **Ours-Stage I** | 0.34 | 0.23 | 0.35 | 0.71 | 0.71 | 0.92 | 1.52 | 0.67 | 0.67 | 0.92 | 1.49 |
| **Ours-Stage II** | **0.57** | **0.43** | **0.47** | 0.91 | 0.88 | 0.95 | 0.98 | 0.86 | 0.87 | 0.95 | 1.04 |

DorsalNet also suggest that accurately estimating dense optical flow is a challenging task, requiring specific design considerations to address complex and long-range spatial interactions, large jumps, and boundary effects, the complexities of which are not adequately captured by simple mechanisms.

## 4 Discussion and Conclusion

DNNs have achieved impressive performance in various vision tasks, and their ability to explain the HVS is an active area of research. Recent studies have employed DNNs to model and understand the neural mechanism of visual motion. For instance, Rideaux et al.[19, 20] and Nakamura and Gomi [21] used multilayer feedforward networks, while Storrs et al. [22] used a predictive coding network (PredNet), to model biological visual motion processing, and found similarities to neurophysiological data. De Jong et al.[68] found that the spatiotemporal frequency tuning properties of some units in FlowNet resemble those found in mammalian neurons. DorsalNet[23] uses first-person perspective video stimuli to train a 3D ResNet model to predict self-motion parameters, which helps model recapitulate the neural representation of dorsal visual stream.

With a similar goal in mind, we simplified the biological motion process pipeline and proposed a two-stage architecture that models the complete pathway from images, through neural representations, to the perceptual response. Through end-to-end training using a wide range of datasets, our model generalizes well from simple stimuli to complex natural scenes and partially captures important characteristics of motion-processing neurons, including a change in spatiotemporal tuning from V1 to MT areas. To model the motion integration function, we introduced a novel recurrent process based on the attention mechanism. This process successfully explains a wide range of physiological findings (e.g., a change in the population of component and pattern cells from V1 to MT) and psychophysical findings (e.g., global motion pooling). It also improves the partial correlation with human psychophysical response. The success of the attention mechanism in motion integration could be attributed to its similarity to the human visual grouping mechanism[69] or similar feature grouping/binding that may be accomplished using a top-down attentional selection mechanism[70].

While we show that the attention-based recurrent network is a promising computational model of human visual motion grouping and segmentation, how its complex architecture (where all responses can be influenced by all other responses) is actually implemented in the human brain remains an open question. Another limitation of our current model is that it does not process several important abilities of human motion perception, including non-Fourier (second-order) motion detection [61], and motion integration sensitive to surface layout[71]. Our model does not take into account the benefits of biological processing, such as energy efficiency.

In conclusion, combining classical motion energy and advanced deep learning technology is a promising approach to bridge the gap between human and DNN motion perception systems. Our proposed architecture and recurrent process offer insights into the underlying mechanisms of motion perception and open up new avenues for future research.

## Acknowledgement

This work was supported in part by JST, the establishment of university fellowships towards the creation of science technology innovation, Grant Number JPMJFS2123; in part by the JSPS Grants-in-Aid for Scientific Research (KAKENHI), Grant Numbers JP20H00603 and JP20H05957.

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
