# OpenReview forum: "Modeling Human Visual Motion Processing with Trainable Motion Energy Sensing and a Self-attention Network"
_NeurIPS.cc/2023/Conference — NeurIPS 2023 poster_

### Official Review · Reviewer_QXFT · 2023-07-05

**Soundness:** 4 excellent
**Presentation:** 4 excellent
**Contribution:** 3 good
**Rating:** 7
**Confidence:** 2

**Summary:**

In this paper, the author tries to build up a visual system like human eyes. In this study, the authors propose a two-stage model that combines trainable motion energy sensing with a recurrent self-attention network to capture the computations in V1-MT, the core structure for motion perception in the biological visual system. The model's unit responses are similar to mammalian neural recordings regarding motion pooling and speed tuning and replicate human responses to various stimuli. The model outperforms several state-of-the-art computer vision models in explaining human responses that deviate from the ground truth.

**Strengths:**

1. This paper has a deep exploration of the human visual system based on on-the-shelf computer vision modules.The idea of using two stage framework to build up this system is insightful.
2. The design of this paper is intuitive and this framework achieves reasonable results.



**Weaknesses:**

1. In this paper, the authors do not provide much visual evidence to prove their system.
2. In some experiments, their method is still worse than state-of-the-art

**Questions:**

None

**Limitations:**

Please refer to weaknesses

---

> ### Author Rebuttal · Authors · 2023-08-05
>
>
> We genuinely appreciate the time and effort of the reviewer. We respond to the concerns pointed out by the reviewer as follows.
>
> ----
>
> **Do not provide much visual evidence to prove their system**:
>
> We entirely agree that for a topic as intricate as motion perception, dynamic visual demonstrations would offer a much more comprehensive and understandable insight than text alone. Unfortunately, due to page constraints in the conference proceedings, it was impractical to provide a thorough visual display of the main content of the paper. To supplement this, we have provided a wealth of visual evidence in the supplementary materials. These include multiple demonstration videos and the results produced by our model. We've added hyperlinks for your convenience, and we trust that these materials will give a more in-depth understanding of our work and its significance.
>
> **In some experiments, their method is still worse than state-of-the-art**:
>
> We recognize that our model's optical flow prediction performance may not be as high as other SOTA models. However, it's important to note that our main goal is to model human motion perception. Therefore, we prioritize biological plausibility over pure performance.
>
> The SOTA models are intentionally designed to achieve the best performance in matching ground truth (GT) data. In contrast, we incorporate additional constraints on our current models to trade off biological plausibility, such as motion energy computation. Hence, it is not appropriate to rely solely on a simple correlation to GT as an index for comparing the capabilities of our model with other SOTA models.
>
> Regarding the comparison with SOTA models, our model is comparable to SOTA CV models when the object is the human response. Table I indicate that some SOTA models might demonstrate a higher correlation with human responses. However, as another study [ref] points out, there's a substantial inherent correlation between the ground truth and human responses. As a result, any model trained to fit the ground truth must consider if the high correlation with human responses is merely a result of fitting the ground truth well. This led us to employ a partial correlation metric, which evaluates the correlation between the model and human responses while controlling for the impact of the ground truth. More intuitively, we exclude the covariance between GT and the model prediction as well as between the GT and human response. Therefore, the partial correlation we used is only related to how strong model prediction could explain the variance in human response, which turns out to be a pure correlation between humans and the model. We believe this offers a more accurate reflection of the model's performance on human response.
>
> When we refer to the partial correlation, our model outperforms the SOTA models in Table I. The likely reason is that the current SOTA models in the computer vision field are primarily designed to match the ground truth as closely as possible, which is not the objective of this study. Instead, we aim to consider a wider range of factors associated with the human visual motion system when developing the model. The data in Table I substantiates that when we eliminate the effect of the Ground Truth, our model tends to align more closely with human responses in complex natural scenes.
>
>
>
>
>
> ---
>
> We trust these clarifications address the reviewer's concerns and offer a more transparent understanding of our work.
> Thanks again for the reviewer's constructive contribution to our work.
>
> Best,
> Authors
>
>
> [ref]
> - Yang, Y. H., Fukiage, T., Sun, Z., & Nishida, S. Y. (2023). Psychophysical measurement of perceived motion flow of naturalistic scenes. Available at SSRN 4414877

---

### Official Review · Reviewer_rMjX · 2023-07-05

**Soundness:** 3 good
**Presentation:** 3 good
**Contribution:** 3 good
**Rating:** 7
**Confidence:** 4

**Summary:**

This paper proposes an image-computable model of human motion perception, bridging the gap between biological computation and CV models. The proposed model contains a two-stage approach that combines trainable motion energy sensing with a recurrent self-attention network for adaptive motion integration and separation. The similarity of the proposed model to human visual motion processing is demonstrated by computer neurophysiology experiments and psychophysics experiments.

**Strengths:**

S1. This study applies DNNs to construct a model of human motion perception that extracts informative motion flows for a wide range of inputs. Specifically, a two-stage model is proposed that simply mimics mammalian V1 and MT functions, respectively.
S2. The two-stage model’s neurons exhibit direction and speed tunings similar to those observed in mammalian physiological recordings in V1 and MT.
S3. Human-like responses are generated to several traditional motion stimuli and illusions, including the global motion pool and the barbershop illusion, showing good generalization from texture-free stimuli (e.g., drifting Garbo) to complex natural scenes.
S4. This paper is clear and well-written overall.

**Weaknesses:**

W1: Although the primary purpose of this paper is to better model human motion perception, there is a gap between the two-stage model and state-of-the-art models in the CV field regarding the optical flow prediction performance, which needs more explanations/analyses.
W2: The proposed two-stage model is relatively complicated, but this study lacks complexity analysis for each part. Considering the low energy consumption property of the human brain, energy efficiency may be another advantage of this model compared with other SOTA models.

**Questions:**

See the above statements and the following questions.
Q1: As this study just uses cosine similarity, is the phrase “self-attention” suitable? Maybe it’s better to use a term from biological computation mechanisms.
Q2: The meaning of some items in Equation 3 is not clear.
Q3: What’s the origin of F in Fig.1E? Is it equal to E? Because the whole procedure is quite complicated, it needs clearer explanations/descriptions.
Q4: What’s the meaning of “Stage II-1/2/3/4” in Fig.3 or Fig.5?
Q5: Mixed usage of “two-stage”, “two-stages” and “two stages”. Please be consistent.
Q6: In Table 1 caption, “speed, and direction” should be “direction, and speed”?

**Limitations:**

This paper has discussed its limitations in modeling human motion perception and its potential influence on future research. Maybe it’s better to add some limitations regarding to its performance on optical flow prediction, compared to other SOTA models.

---

> ### Author Rebuttal · Authors · 2023-08-06
>
>
> Thank you for the reviewer's comprehensive comments and for recognizing the value of our work.
>
> ---
>
> **The gap between our two-stage model and SOTA models in the CV field regarding optical flow prediction performance:**
>
> We recognize that our model's optical flow prediction performance may not be as high as other SOTA models. However, it's important to note that our primary goal is to model human motion perception. Therefore, we need to trade off biological plausibility over pure performance.
>
> Our model integrates classical motion energy computation with a graph connection and attention mechanism to solve dense optical flow in real-world scenarios effectively. In the design stage, using many stacked CNN layers instead of motion energy computation might enhance performance but would obscure each layer's specific functionality, deviating from our initial intention. Instead, our model is designed for clarity of function: the first layer extracts local energy, while the second layer manages motion integration and segregation.
> CV models are designed to match Ground Truth (GT) data best, but our approach adds extra constraints to maintain a balance of biological plausibility. Therefore, a simple correlation to GT shouldn't be the sole comparison index between our model and other SOTA models.
>
> Regarding Table I, our model is comparable to SOTA CV models when the object is the human response. Some SOTA models demonstrate a higher correlation with human responses. However, as a recent study (Yang et al. 2023) points out, there's a substantial inherent correlation between GT and human responses. As a result, any model trained to fit the ground truth must consider if the high correlation with human responses is merely a result of fitting the ground truth well. This led us to employ a partial correlation metric, which evaluates the correlation between the model and human responses while controlling for the impact of the GT. When evaluated using the partial correlation, our model outperforms the SOTA models with a considerable gap in Table I. Our preliminary analysis suggests that our model best explains human-perceived flow in naturalistic scenes because the attention network can do something similar to vector decomposition (Johansson, 1973).
>
> **The proposed two-stage model is relatively complicated and lacks complexity analysis for each part:**
>
> Our current focus is to make a computational model similar to humans and competitive with SOTA CV models without seriously prioritizing the complexity/efficiency of computation.
>
> The choice of our framework is based on its algorithmic affinity to complex neural computations in the human brain. The two-stage concept is a simplified yet representative version of the real-world structure of the human brain's motion processing system. Considering the complex topology and dynamic temporal characteristics of brain neurons, we have incorporated graph structures and recurrent neural networks to deal with the motion integration tasks, a function of the MT regions. With a total of 14.7 million trainable parameters, our model remains leaner compared to generic CNN backbones.
>
> Our study suggests what kinds of computational mechanisms are necessary to explain human perception. It is possible that the neural implementation for this computation is different and more efficient.
>
> **Questions**:
>
> - Q1: As this study just uses cosine similarity, is the phrase "self-attention" suitable?
>
> Thanks for your suggestion. According to our understanding, the "self-attention/ attention mechanism" in computer vision and deep learning fields is a kind of calculating the similarity between each pair of spatial location/ temporal step across the 2-D feature map / 1-D sequence. Most works, such as current transformer architecture, use dop-product distance as the distance metric. The cosine similarity used in this work is also a mathematical distance metric. Using different distance metrics wouldn't deviate from the same concept of self-attention. The cos similarity can be decomposed into dot-product with a normalization factor, and it could normalize numeric value into [-1,1]. In our experiment, we found that this feature helps the numeric stability of the model when training.
>
> - Q2: The meaning of some items in Equation 3 is not clear.
>
> Apologies for the confusing mathematical symbols. The symbol $\Re$ and $\Im$ denote acquiring the Real and Imaginary parts from a complex value, respectively.
> We will replace that with **Im[.]** and **RE[.]** for a better understanding. In addition, $*$ means the convolution operation.
>
> - Q3: What's the origin of F in Fig.1E? Is it equal to E? needs clearer explanations/descriptions.
>
> If we refer to Fig.1 D, the RMIB block corresponds to Fig 1. In the initial recurrent stage, the red dot (F) and blue dot (E) both originate from the same motion energy in Stage I. However, in subsequent recurrent stages, 'F' and 'E' begin to diverge. 'F' represents the guiding attention for global motion integration, whereas 'E' signifies the motion energy response utilized to decode the optical flow.
> We acknowledge that this process could use more clarification and will enhance the captions to explain the model's workings better. Furthermore, we found a labeling error in Fig 1 legend (in the lower right corner), where the descriptions of the blue and red dots should be switched. These will be corrected in our revised manuscript.
>
> - Q4: What's the meaning of "Stage II-1/2/3/4" in Fig.3 or Fig.5?
>
> Stage II is a recurrent stage, and we analyze the result of each recurrent iteration. Stage II -1/2/3/4 means the result from iteration 1/2/3…
> We will provide details in the revision.
>
> - Q5: Mixed usage of "two-stage", "two-stages" and "two stages".
>
> - Q6: In Table 1 caption, "speed, and direction" should be "direction, and speed"?
>
> Thanks for the suggestion. We will modify it.
>
> ---
>
> We hope our responses sufficiently address the reviewer's concerns.
>
> Best,
>
> Authors

---

### Official Review · Reviewer_gk6R · 2023-07-07

**Soundness:** 4 excellent
**Presentation:** 3 good
**Contribution:** 4 excellent
**Rating:** 8
**Confidence:** 3

**Summary:**

I have read the authors' rebuttal and will maintain my already high rating.

This paper proposes a new model of the dorsal pathway (V1->MT) using a two-stage architecture. The first stage uses spatiotemporal filters tuned by supervised learning, while the second stage uses a dynamic connection between motion detectors based on the similarity of motion responses. The model is able to capture several important aspects of human motion processing, including both neurophysiological properties and psychophysical responses.

**Strengths:**

-The paper is very well written, although there are a few places where it is not clear.

-The model is quite convincing in its fits to the data, both neurophysiological and psychophysical.

-The models fits to data are compared to 8 different SOTA models of optical flow, and when the correlation to the ground truth is factored out, the model is superior to all others (by having a high correlation to the ground truth motion, the models will necessarily	have a high correlation with the human data). This is an important test.

-The model can not only account for responses to low-level stimuli (drifting gabors), but can also give convincing responses to natural and artificial movies. I am not an expert in motion processing, but to my knowledge, this is the first model to do this. Perhaps other reviewers will know of other models with this capability.

-The model is a sophisticated one, using a dynamic graph construction that connects the motion detectors in stage 2, integrating the motion detectors in stage 1 based on similarity of response. In this way, the model can capture global motion, solving the aperture problem.

**Weaknesses:**

-The model’s sophistication is also a weakness: The dynamic graph construction assumes all responses can be influenced by all other responses, depending on the response similarity. That is, dynamically, any response integrator can be connected to any other, no matter how far away spatially. This is what allows the network to integrate the local motion into global motion. How this could be implemented neurally is very unclear. However, this is an argument from lack of imagination. Also, the paper acknowledges this limitation.

-Some of the presentation is unclear/unconventional: For example, figure captions don’t label all components of the figure (e.g., the caption to Figure 5 only describes 5D), although they may be discussed in the text. It is not always clear what is being shown in the Figures, which are relatively complex.

**Questions:**

1. What do the script symbols in Equation 3 represent? They aren’t defined.


Line 67: mode’s -> model’s

123: ability to capture local motion only ->
        ability to capture only local motion.

132: process -> processing

bottom of page 4: At this point, it would be great to give an intuitive description of what this adjacency matrix represents. It looks to me like this strongly and globally  connects neurons with similar motion responses, which would make intuitive sense, but since it is dynamically computed, it isn’t obvious how this would be implemented neurally.

184: our primary lies -> our primary goal lies

It’s unclear what’s being shown in Figure 3D, especially the graphs in the first row on the right (is this still part of D?). Presumably, lines 227-231 are describing what this is, but I don’t understand the explanation.

line 251 refers to figures 3A and 4A. Are both supposed to be 4A?

Also, panel E should be to the left of panel F.

Figure 4B middle panels. What are these depicting?

**Limitations:**

Barber pole illusion is not quite captured properly (Figure 4D), although the end points are.

The neural implementation of this model is unclear. This is mentioned as a limitation.

---

> ### Author Rebuttal · Authors · 2023-08-05
>
>
>
> Thanks for the reviewer's appreciative comments regarding our work, and I'd like to address the reviewer's concerns as follows.
>
> ---
>
> **On the neural implementation of dynamic graph construction**:
>
> We admit that the specific neural implementation of the attention mechanism is still unclear. This is a complex neuroscientific issue and one of the limitations we acknowledge in our paper. However, numerous psychophysical findings suggest the presence of global motion integration capabilities. For instance, human visual motion computation exhibits substantial interactions over vast spatial separations (refer to Maruya, K., Holcombe, A. O., & Nishida, S., 2013). To simulate this, we required a function capable of flexibly integrating spatial information, which is what the attention mechanism and graph topology effectively provide.
> As the reviewer correctly pointed out, the current version of our model assumes, dynamically, any response integrator can be connected to any other, no matter how far away spatially. This may sound physiologically plausible. However, we could incorporate position embedding to mirror retinal topological information when calculating the adjacency matrix. Consequently, as illustrated in our supplementary material (Fig. 1.2.3). The visualized connections primarily focus on the local area of the selected regions.
> However, considering that human visual motion computation exhibits substantial interactions over 100 deg (see, e.g., Maruya, Holcombe & Nishida, 2013), designing biologically plausible position embedding is not an easy task. It is possible that the neural implementation for this computation is different and more efficient, and we will acknowledge this limitation in the manuscript.
>
>
> **Addressing presentation clarity**:
>
> We apologize for any confusion stemming from our figures and captions. Our revised manuscript aims to enhance figure clarity and provide more detailed captions to explain each component thoroughly.
>
> **Questions**:
>
> - **Q1: What do the script symbols in Equation 3 represent?**
>
> A1: We apologize for the lack of explanation. The symbols $\Re$ and $\Im$ extract the real and imaginary parts of a complex number, while $*$ denotes convolution operations. We will add these definitions in our revised manuscript for clarity.
>
> - **Q2: some typo errors**
>
> A2: Thank you very much for the careful check and correction. We will correct them in our revised manuscript.
>
> - **Q3: bottom of page 4: give an intuitive description of what this adjacency matrix represents**
>
> A3: Thanks for your suggestion. Intuitively, this adjacency matrix represents the neuron's affinity or connectivity within the space. We will provide a more in-depth explanation in the revised manuscript.
>
> - **Q4. It's unclear what's being shown in Figure 3D, especially the graphs in the first row on the right (is this still part of D?).**
>
> Firstly, considering the clarity, we consider relocating the graphs in the first row to the supplementary material or appendix.
>
>  Fig. 3D and its accompanying content are designed to investigate the spectral properties of the units in our model. To accomplish this, we subjected the model to a combination of drifting-Gabor stimuli with varying spatiotemporal frequency components, which we can interpret as testing the spectral receptive field of each unit.
>
> To analyze these receptive fields, we employed a 2D Gaussian profile fitting. This mathematical approach provides key parameters such as the central location and oblique angle of the fitted profile. These parameters allow for a quantitative examination of the spectral field distribution for each neuron. The upper-right section of Fig 3D demonstrates the distribution of these fitted Gaussian profiles. Each small dot corresponds to the central location of a unit's receptive field in the frequency domain. The slanted bar associated with each dot signifies the oblique angle of the spectral receptive field for the corresponding neuron. The length of the bars represents the eccentricity of the receptive field. This collection of oblique angles is further visualized as a bar plot in Figure 4E.
>
> However, we want to stress that Fig 3D primarily serves to introduce Figure 4E. It does not aim to advance any argument. Please refer to the supplementary material (Line 117) for more details.
>
> - **Q5. line 251 refers to figures 3A and 4A...**
>
> Thanks for your correction! In line 251, we changed 'Fig.3(A)' to 'Fig. 4'.
> We will also adjust the arrangement of panels E and F.
>
> - **Q6. Figure 4B middle panels...**
>
> The middle panels of Figure 4B aim to represent the neuron's connectivity by visualizing the adjacency matrix. The heat map indicates the neuron's connectivity (cosine similarity in the adjacency matrix) from a selected local region to other regions. The warmer the color, the higher the similarity. The two figures depict how units with high activity establish long-distance connections to resolve the aperture problem when subjected to Gabor (ambiguous motion) stimuli. In contrast, the plaid stimuli (determined motion) suppress these long-distance connections. We will provide details in captions.
>
> *More details*:
> Fig. 4 (B) compares spatial motion integration between 1D Gabor motion (left) and 2D plaid motion (right). Humans are able to perceive global downward motion only in the former case. In the latter case, local integration of motion signals takes priority over global integration. Once the local ambiguity is resolved, the global integration process is suppressed. Our model can predict such adaptive motion pooling in human visual processing. (K. Amano et al., 2009)
>
> ---
>
> We hope our responses address the reviewer's concerns and appreciate the reviewer's constructive feedback.
>
> Best,
> Authors

---

### Official Review · Reviewer_n3Ed · 2023-07-08

**Soundness:** 3 good
**Presentation:** 2 fair
**Contribution:** 3 good
**Rating:** 6
**Confidence:** 3

**Summary:**

This paper proposes a novel model of motion analysis. It takes inspiration from biological motion processing to try and solve the aperture problem, leveraging a combination of biologically inspired constrained structure with learned parameters. This produces strong partial correlations of motion components in empirical tests, but also interesting emergent phenomenon when investigating the distribution of unit responses between the different network components.

**Strengths:**

- This paper does a good job of anchoring model development with a strong motivation and justifications based on biological understanding of motion processing. This is not an easy task, and the paper does a good job of finding compromise between the performance-driven approach of data fitting with the desire for an explanatory model that provides insight into why a certain performance or behaviour is achieved.

- The paper is quite dense with a wide range of ways to explore the problem and the model. This is both a strength and a weakness, as it sometimes makes the paper difficult to follow, but the range of analysis is impressive.

**Weaknesses:**

- Missing reference: Tsotsos et al., "Attending to visual motion", CVIU 2005
  -- This paper provides a biologically motivated model of motion including attentional effects and structure informed by our understanding of visual areas V1, MT, MST, and 7a in the primate visual system.

- Figure 1 is rather confusing; there's a lot going on, and I found flow of information rather unclear. If possible I would recommend breaking it up into separate figures or finding a way to show how the different components connect in a clearer manner. For example, is C a sub-component of A, or just some example receptive fields? If the latter, why is it part of the network diagram? Similarly, in the text F is given as having dimensions HWx256 (Line 140), but in the RMIB section of Figure 1 it looks like it is supposed to have HxWxC dimensions. Is this a different F? This was not clear.

- The partial correlation metric should be clearly defined; I did not see a definition for this in the paper. It would be better to have more discussion of the aspects that the model does not excel at compared to competing models (e.g. FlowFormer for v.s. Human and RAFT for vs. GT in Table 1).

**Questions:**

- In the caption for Table 1 it states that uv = Cartesian space, dir = speed, and spd = direction. I assume dir should be direction and spd should be speed?

**Limitations:**

Limitations seem to be adequately discussed.

---

> ### Author Rebuttal · Authors · 2023-08-05
>
>
> Thanks for the constructive feedback from the reviewer.
>
>  -----
>
> **Missing reference**:
>
> We will include the recommended reference: Tsotsos et al., "Attending to visual motion," CVIU 2005, in the introduction and discussion part.
>
> **Clarification of Figure 1**:
>
> We understand that Figure 1 seems overcrowded and potentially confusing. We will work on refining it for improved clarity and coherence.
>
> As for the reviewer's specific questions, 'Fig. 1C' is indeed a sub-component of 'Fig. 1A'. We designed the model such that there are 256 different motion energy units in Stage I and 'Fig. 1C' visualizes one of these units. This unit consists of a quadrature pair of spatial and temporal filters, forming a spatiotemporal slanted receptive field. We will make this relationship clearer in the revised manuscript.
>
> As for the dimensions of F, we apologize for the confusion. The correct description should be $F \in \mathbb{R}^{H×W×C}$ where $C = 256.$ (Line 140)
>
> *Here we provide some additional context:*
>
> In our model, C represents the number of channels in a given feature map, which in this case is 256. Hence, we have $H×W×256$. This is equivalent to $HW×256$; the only difference lies in the shape in which we represent the same quantity. When we say $F \in \mathbb{R}^{H×W×C}$, we refer to a 3D space. On the other hand, $HW×C$ represents the same set of elements but flattened along one dimension—akin to reshaping a 2D image into a 1D sequence. This change in representation is primarily for the convenience of matrix product calculation to get the adjacency matrix. While this convention is commonly used when defining tensor shapes in the context of neural networks, we acknowledge that it may not be mathematically rigorous.
>
> We will include these clarifications in our revised manuscript.
>
>
> **The partial correlation metric should be clearly defined; I did not see a definition for this in the paper. It would be better to have more discussion of the aspects that the model does not excel at compared to competing models (e.g., FlowFormer for v.s. Human and RAFT for vs. GT in Table 1).**
>
> The partial correlation we used is defined as:
>
> $$
> \rho_{\text {model }}=r_{\text {resp model } \cdot G T}=
> \frac{
> r_{\text {resp model }}-r_{\text {resp } G T}\cdot r_{\text {model } G T}} { \sqrt{1-r^2_{\text {resp } G T}} \sqrt{1-r^2_{\text {model } G T}}}
> $$
>
> where $r$ is the Pearson correlation. Due to page limitations, we briefly mentioned the partial correlation from Line 283 but did not detail the specific equation definition. To ensure clarity, we will provide the definition of partial correlation in our supplementary material.
>
> Here, we would like to explain the reason for using partial correlation, combining the question of "*FlowFormer for v.s. Human and RAFT for vs. GT.*"
>
> We recognize that our model's optical flow prediction performance may not be as high as other SOTA models. However, it's important to note that our main goal is to model human motion perception. Therefore, we prioritize biological plausibility over pure performance.
>
> The SOTA models are intentionally designed to achieve the best performance in matching ground truth (GT) data. In contrast, we incorporate additional constraints on our current models to trade off biological plausibility, such as motion energy computation. Hence, it is not appropriate to rely solely on a simple correlation to GT as an index for comparing the capabilities of our model with other SOTA models.
>
>
> On the other hand, we believe that considering the correspondence of the model prediction to human response provides a fairer index for comparing models. However, previous psychophysical studies have highlighted that human response is inherently highly correlated with GT [ref]. This correlation might lead to the observation that any models trained to fit the GT are also highly related to human response, regardless of the model design. Following the previous study's strategy, to address this potential confusion between human-model correspondence (true evaluation) and human-GT correspondence (confounding), we employ partial correlation, controlling of GT, to purify the true model-human correspondence. More intuitively, we exclude the covariance between GT and the model prediction and also between the GT and human response. Therefore the partial correlation we used is only related to how strong model prediction could explain the variance in human response, which turns out to be a pure correlation between humans and the model.
>
> Based on the partial correlation metric evaluation, our model surpasses the state-of-the-art models listed in Table I. In addition, the pure EPE data proves that our model closely matches human responses in complex natural environments, whereas other state-of-the-art models tend to align more with the ground truth.
>
> We hope the above explanation provides you with a better understanding.
> We will detail these points further in our revised manuscript.
>
>
> **Caption for Table 1:**
>
> Thanks for noticing this mistake. 'dir' should indeed denote 'direction,' and 'spd' should denote 'speed.' We will rewrite the caption to:" uv, dir, spd represent motion components in Cartesian space, direction, and speed, respectively. "
>
>
> [ref]
>
> - Yang, Y. H., Fukiage, T., Sun, Z., & Nishida, S. Y. (2023). Psychophysical measurement of perceived motion flow of naturalistic scenes. Available at SSRN 4414877
>
>  ----
>
> We are appreciative of your time and effort spent reviewing our work and giving us valuable insights.
>
> Best,
> Authors

---

### Official Review · Reviewer_tFBw · 2023-07-21

**Soundness:** 3 good
**Presentation:** 3 good
**Contribution:** 2 fair
**Rating:** 6
**Confidence:** 4

**Summary:**

This paper proposes a new V1-MT model using a normalized Gabor model of V1, followed by a recurrent self-attention stage. It uses dense optic flow as a supervised objective. The authors perform extensive in silico neurophysiology to show the model units qualitatively look like V1 and MT. They also show that this model is a better match to human visual perception on natural scenes than computer-vision models.

**Strengths:**

- Well-written and clear
- Breadth of in silico experiments
- The model makes sense and I’m excited about the idea of segregation vs. integration as an explanation for complex receptive fields in MT

**Weaknesses:**

- Pretty incremental in a crowded field
- Little comparison to SOTA
- Mostly qualitative and little quantification

I really want to like this paper: full disclosure, I’m a big fan of the work of Orban et al. and more recently Cui et al. (2013) that shows that MT cells have complex receptive fields capable of integration and segregation. However, this paper kind of rubbed me the wrong way by ignoring the state-of-the-art in this field: “Despite extensive research in cognitive neuroscience, image-computable models that can extract informative motion flow from natural scenes in a manner consistent with human visual processing have yet to be established”. There are plenty of image-computable models that can extract information relevant to dense optic flow from natural scenes. Everything from the old work of Simoncelli and Heeger to the receptive field models empirically derived from MT of Nishimoto and Gallant (2011) to MotionNet from Rideaux et al. and especially to Mineault et al. (2021), inexplicably uncited in this manuscript despite being published in these very pages and having very similar motivation to this manuscript. The onus is on the manuscript to show us some failings of these previous models and how the model does better according to some axis.

To be fair, the paper could say that some of these networks–especially generic 3d CNNs without multi-scale representations and explicitly organized direction and speed tuning–do not extract quantitative optic flow information *explicitly*. However, consider this thought experiment: if I wanted to read information from a patch of MT, as a psychophysical observer presumably needs to do to solve the tasks in the paper, I wouldn’t have access to a neat organization of direction and speed tuning, either; I would need to do a readout on top of the patch. I don’t buy the premise that the brain needs to form an explicit estimate of dense optic flow at every point in space, e.g. in MT. I think it’s easy enough to turn these old models into ones that estimate dense optic flow using a linear readout, and these should be compared to the proposed model.

The paper spends a lot of time rehashing the same kind of qualitative receptive field exploration that’s a hallmark of this field: tuning curves, component vs. pattern, speed tuning curves, distribution of preferences for direction tuning, barber poles, reverse phi, etc. This is a very crowded field between all the papers from Rideaux, Welchman, Fleming, Mineault, Bakhtiari and Pack: these kinds of demonstrations have been done over and over again. It’s nice to have it in the paper but I really consider these a sanity check, not a finding, especially since this paper bakes in Gabor receptive fields in the first layer: how could they not learn direction selectivity? My advice, speed run through these to give more space for the end of the paper, which is where things get interesting.

Definitely the most interesting thing about the paper, in my view, is Table 1. It shows that the network knows something about how human motion estimation operates that is not captured by CV models. It’s a bit buried in the paper and explained a bit fast. The authors should add ablations to figure out what it is about this network that makes it more brain-like - the Gabors? the normalization? the attention mechanism? I also think they need to add in quantitative comparisons with MotionNet from Rideaux et al. and DorsalNet from Mineault et al. with a linear decoder on top–to be clear, I’d be fine with a linear decoder trained on another task. You could avoid doing a linear decoder by using an RSA-based analysis, or you could use other methods of alignment which are more restrictive than a linear decoder (e.g. from Alex Williams et al.), if you believe this is not an apples-to-apples comparison.

Overall, I think this could be a valuable contribution to the field, but it needs to be very explicit about its specific contribution in light of plentiful previous work, and it needs explicit comparisons to this previous work. I would be happy to accept granted the authors cite and address previous literature and include quantitative comparisons to MotionNet and DorsalNet, provided their model comes out on top either according to the metrics they have in Table 1 and Figure 5 or some other relevant metric they find.

Nitpicks:

- I liked the convention of using red color for trainable parameters, but the authors only use that once and that drop that later. Would recommend doing it consistently throughout the methods.
- Bottom of Page 4: “Top side of Fig. 3” → Should be a reference to figure 1. There are a couple more instances of this, so the authors should go carefully through the manuscript to find if there are more instances of that.
- The model is fairly bespoke but it’s not particularly biologically plausible with its attention mechanism. If the point is to implement recurrence, why not just use a plain transformer with tied weights at every layer instead? To be clear, the authors don’t have to fit this model, just a sentence or two to justify why they picked this architecture rather than something more off-the-shelf.

**Questions:**

-

**Limitations:**

-

---

> ### Author Rebuttal · Authors · 2023-08-06
>
> Many thanks for thoroughly reading our research and comprehensive comments.
> Firstly, we appreciate the reviewer's nitpicks and will make the suggested modifications.
>
> ---
>
> **General Response:**
> - We should clearly state the purpose of our study. From a scientific perspective, we aim to explain a wide range of psychophysical phenomena, including those whose physiological mechanisms are not yet clear, while aligning the model's internal representations with physiological ones. Engineering-wise, we aim to make a human-aligned model that maintains competitive performance with SOTA CV models. In contrast, the purpose of MotionNet and DorsalNet was to explain neural responses to visual motion stimuli. They are good models for this purpose, but they do not give a dense optical flow. We cannot compare the outputs of these models with human motion perception, our model, and CV models.
>
> - We do not believe in an explicit representation of dense optical flow in the brain. However, given that it is possible to reproduce a high-resolution flow from human responses (Yang et al., 2023) and that many motion phenomena include interactions of local motion signals, constructing a model like ours would be useful for a computational understanding of motion perception.
>
> - The reviewer suggested that MotionNet/DorsalNet with a simple linear encoder to compute a dense optical flow might outperform our model. Accurate dense optic flow estimation is a difficult task that requires complex and long-range spatial interactions to tackle large jumps and the boundary effect. FlowNet (Dosovitskiy et al., 2015), the first optical flow model to use a multi-layer stacked CNN, has more than ten times the parameters of MotionNet but still lags behind SOTA models in Table I. The CV SOTA models employ many sophisticated strategies to match GT and better predict human responses. In the attached PDF, we provide additional results obtained with three relevant models: a pre-trained DorsalNet with a linear flow decoder, a general 3D CNN (basic structure for MotionNet and DorsalNet), and FFV1MT (Solari et al.,2015), the only model known to calculate a dense optical flow with simple decoding of the Simoncelli & Heeger V1-MT mechanism. These models fall short in terms of dense optical estimation accuracy or fail to account for the global integration of local motions. This outcome suggests that a simple linear decoder cannot handle dense optical flow, particularly in complex real scenes. Furthermore, in addition to flexible spatial pooling and accurate estimation of dense optical flow, our model displays consistent human-perceived Fourier motion in missing fundamental stimuli by motion energy computation simulating the V1 neurons. Comparison with other models shows how difficult it is to achieve such diverse capabilities **simultaneously** in a single model.
>
> - We understand the concerns regarding our model being 'over-designed' compared to general 3-D CNNs, and we provide more explanation in the appendix.
>
> - It is not a simple sanity check to show that our model achieves internal representations similar to MT concerning component/pattern motion and spectral receptive field, since stage II of our model is a complex recurrent self-attention network. RSA-based model comparison is an interesting idea to test in the future, but in this study, we do not propose our model outperforms MotionNet and DorsalNet in explanation of MT/MST responses to specific types of motion stimuli used previously in the labs. Our preliminary analysis suggests that our model best explains human-perceived flow in naturalistic scenes because the attention network can do something similar to vector decomposition (Johansson, 1973).
>
> - The neural implementation of the attention mechanism in our model design might look too complicated and biologically implausible. However, substantial psychophysical evidence necessitated a function for flexible spatial information integration, which the attention mechanism and graph structure capably deliver. Simple convolution stacking may be able to yield similar outcomes but will make the model more complex and biologically uninterpretable.
>
> We appreciate the reviewer's insightful feedback. In the revised manuscript, we will clarify how our study differs from previous studies, especially DorsalNet and MotionNet, and make modifications based on the reviewer's minor comments.
>
> Best, Authors
>
>  ---
>
> **Appendix:**
>
> To model a function that allows flexible integration of spatial information, we proposed the attention mechanism and graph structure to accomplish this effectively. Stacking simple convolutions could achieve similar results but is restrictive, setting an upper limit on long-range spatial relationships. For instance, the large 2K resolution Sintel dataset we used for comparison with humans requires hundreds of (3x3) convolution layers for single-time information propagation (Zhao et al., 2017). Stacking such many layers is unreasonable, leading to model bloating, limited long-distance interaction, and convergence issues. In contrast, our model enables graph topology, and the attention mechanism allows global interaction with one-layer operation. MotionNet's simplicity is due to its limited testing stimulus of 32x32, which is manageable with a few CNN layers. However, this approach isn't suitable for larger images due to physical long-distance information limits. Plus, with too many CNN layers, it becomes challenging to comprehend the exact functionality of each layer.  Instead, our model is designed for clarity of function: the first layer extracts local energy, while the second layer manages motion integration and segregation.
> Our use of the attention mechanism also offers the possibility of understanding motion integration by visualizing the adjacency matrix, as shown in Appendix Fig. 1 of our supplementary materials. This level of understanding would not be achievable by simply stacking CNN layers.

---

> > ### Comment · Reviewer_tFBw · 2023-08-13
> >
> > Very cool! Glad you followed through on the evaluation. I've bumped up my rating.

---

### Author Rebuttal · Authors · 2023-08-07

We provided more comparison results in the attached PDF.

---

### Decision · Program_Chairs · 2023-09-21

**Decision:**

Accept (poster)

**Comment:**

There was general agreement among the reviewers that the proposed model of the dorsal stream adds value to an already crowded field of computational models. In particular, the reviewers noted that the model appears sophisticated compared to prior models and that the breadth of in-sillico experiments presented is to be lauded. Reviewers asked for clarifications and additional experiments that were all answered in the rebuttal. The AC recommends this paper be accepted.